# Crosstalk between Hypoxia and Extracellular Matrix in the Tumor Microenvironment in Breast Cancer

**DOI:** 10.3390/genes13091585

**Published:** 2022-09-03

**Authors:** Yasmin Dekker, Sylvia E. Le Dévédec, Erik H. J. Danen, Qiuyu Liu

**Affiliations:** 1Leiden Academic Centre for Drug Research, Leiden University, 2333 CC Leiden, The Netherlands; 2School of Life Sciences, Beijing University of Chinese Medicine, Beijing 100102, China

**Keywords:** tumor microenvironment, cell metabolism, mechanotransduction, hypoxia, extracellular matrix, YAP/TAZ, breast cancer

## Abstract

Even though breast cancer is the most diagnosed cancer among women, treatments are not always successful in preventing its progression. Recent studies suggest that hypoxia and the extracellular matrix (ECM) are important in altering cell metabolism and tumor metastasis. Therefore, the aim of this review is to study the crosstalk between hypoxia and the ECM and to assess their impact on breast cancer progression. The findings indicate that hypoxic signaling engages multiple mechanisms that directly contribute to ECM remodeling, ultimately increasing breast cancer aggressiveness. Second, hypoxia and the ECM cooperate to alter different aspects of cell metabolism. They mutually enhance aerobic glycolysis through upregulation of glucose transport, glycolytic enzymes, and by regulating intracellular pH. Both alter lipid and amino acid metabolism by stimulating lipid and amino acid uptake and synthesis, thereby providing the tumor with additional energy for growth and metastasis. Third, YAP/TAZ signaling is not merely regulated by the tumor microenvironment and cell metabolism, but it also regulates it primarily through its target c-Myc. Taken together, this review provides a better understanding of the crosstalk between hypoxia and the ECM in breast cancer. Additionally, it points to a role for the YAP/TAZ mechanotransduction pathway as an important link between hypoxia and the ECM in the tumor microenvironment, driving breast cancer progression.

## 1. Introduction

Breast cancer is the most commonly diagnosed cancer among women, with an estimated 2.3 million new cases worldwide in 2020 [1,2]. Classification of breast cancer is based on the expression of the following three key receptors: the estrogen receptor (ER), the progesterone receptor (PR), and the human epidermal growth factor receptor 2 (HER2) [3]. Current treatments, including chemotherapy, radiation, surgical resection, and targeted hormone therapy, prolong the overall survival and vastly reduce the mortality of this disease [4]. However, mortality remains high when metastasis and drug resistance develop, especially in triple negative breast cancer (TNBC) which lacks expression of ER, PR, and HER2. Therefore, breast cancer is still one of the leading causes of cancer death in women [5]. Although the mechanisms of breast cancer progression have been widely investigated, further research is still necessary to find novel drug targets and improve the treatment of these patients.

A rather promising treatment strategy involves targeting cancer cell metabolism [6]. A deregulated cellular metabolism is regarded as one of the hallmarks of aggressive cancers [7]. More aggressive tumors consume less oxygen due to their tendency to undergo glycolysis even under non-hypoxic conditions, also known as aerobic glycolysis or the Warburg effect [8]. Even though glycolysis yields less ATP per molecule of glucose than oxidative phosphorylation, it produces ATP more rapidly. As a result, the tumor can meet its high energy demands to fuel processes, such as growth, invasion, migration, and matrix degradation. Additionally, the high amounts of lactate released by the tumor lower the extracellular pH, allowing degradation of the ECM basement membrane, tumor invasion, and suppression of immune effectors (reviewed in [9]). The tumor starves the neighboring cells of nutrients as well, thereby creating additional space for the tumor to grow. The reduced oxygen consumption also grants the tumor a growth advantage in hypoxic conditions [9].

The tumor microenvironment (TME) plays a notable role in cancer progression. It includes pH and oxygen levels, the extracellular matrix (ECM), connective tissue, infiltrating immune cells, and the vasculature of the tumor. Hypoxia might have a vital role in cancer metabolism and metastasis [10]. Hypoxic regions are present in many solid tumors, as oxygen availability is limited due to rapid tumor growth and disordered vasculature. In non-malignant cells, hypoxia typically leads to cell cycle arrest and apoptosis. Conversely, tumor cells are able to acclimatize and survive in these conditions, allowing for tumor growth, invasion, and metastasis [11]. Furthermore, the importance of tissue mechanics in cancer progression has become clearer over the last several decades. Interaction between the ECM and the tumor activates key signaling pathways that promote tumor proliferation, invasion, and metastasis. This notably influences many tumors as the ECM can comprise up to 60% of the tumor mass [12]. Additionally, biophysical properties of the ECM, such as the stiffness, have been shown to influence tumor progression as well [13].

Although hypoxia and the ECM were initially thought to independently influence cell metabolism and metastasis, emerging data indicate a close connection. The aim of this review is to discuss recent evidence pointing to mechanisms underlying crosstalk between the ECM and hypoxia in the TME, as a driving force impacting on breast cancer progression.

## 2. Hypoxia in Breast Cancer

To date, a number of studies have identified a link between hypoxia, metastasis initiation, and drug resistance [14,15,16]. Almost 30 years ago, Young et al. already showed that hypoxia in solid tumors affects the behavior of cancer cells. In their experiment, the metastatic efficiency of cancer cells subjected to hypoxia (<0.1% O_2_), followed by reoxygenation, was two-fold that of cells in well-oxygenated regions [17]. Moreover, data collected from at least 125 clinical studies reviewed by Vaupel et al. suggest that hypoxia is prognostic for tumor progression in various cancers, thereby again supporting this hypothesis [18]. Hypoxia usually occurs at around 100 µm from the nearest functional blood vessel. In breast cancer, oxygen tension ranges from 2.5 to 28 mm of mercury (Hg) which is in stark contrast with the 65 mm Hg in normal human breast tissue [18]. Oxygen tension values of less than 10 mm Hg are linked to an increasing risk of metastasis, thus, indicating that hypoxic breast cancers show more aggressive behavior [19]. While the ability of hypoxia to influence the metastatic cascade is recognized, the mechanism behind it is incompletely understood. The most extensively studied mechanism of how cancer cells adapt to hypoxic stress involves hypoxia-inducible factors (HIFs).

### 2.1. Regulation of HIF Pathways

The HIFs are transcription factors comprising an oxygen sensitive α subunit and constitutively expressed β subunit that act as heterodimers [20,21]. There are three different members of the HIF family in humans: HIF-1, HIF-2, and HIF-3. While the HIF-1α subunit is the most prominent of the HIFs in hypoxia, Hu et al. showed that HIF-2α also regulates hypoxia-responsive genes and is not restricted to endothelial cell-specific gene expression as believed earlier [22]. Even though HIF-2α is structurally similar to HIF-1α, as they share a 48% identical amino acid sequence, many studies utilizing various approaches show that they have unique roles and gene targets in both normal and cancer cells [23,24,25]. Conversely, HIF-3α has a different amino acid sequence from both HIF-1α and HIF-2α. More to the point, HIF-3α is often deemed a negative regulator of HIF-1α and HIF-2α target genes [26]. Indeed, HIF-3α owes this status to its adverse effect on gene expression due to competitive binding between the HIFs to the transcriptional response elements of target genes. This review will primarily focus on HIF-1α as it is the most prominent HIF during hypoxia.

Under normoxic circumstances, HIF-1α is ubiquitinated by binding to the von Hippel-Lindau tumor suppressor protein (pVHL) and subsequently degraded. Conversely, HIF-1α ubiquitylation is inhibited in hypoxic regions, as the binding to pVHL is regulated by oxygen-dependent prolyl hydroxylation. As a result, HIF-1α accumulates and forms a heterodimer with HIF-1β. The HIF-1 then translocates to the nucleus and activates up to 200 genes, including genes implicated in angiogenesis and cell metabolism [27,28].

Besides reduced oxygen levels, numerous oncogenic modifications in cancer cells can enhance HIF expression as well. These modifications take place in pathways involved in HIF synthesis, stability, or transactivation. Examples observed in several studies are loss of function of pVHL, p53, or factor inhibiting HIF-1 (FIH-1), and activation of the phosphatidyl inositol-4,5-bisphosphate-3-kinase (PI3K)-protein kinase B (PKB)/AKT pathway [29,30,31]. Mechanistically, FIH-1 blocks the transactivation of HIF-1α via oxygen-dependent hydroxylation and, thereby, stops the binding of HIF-1α, HIF-1β, and co-activator CBP/p300 [32]. Here, P53 acts similar to pVHL, as it regulates the stabilization of HIF-1α by promoting Mdm2-mediated ubiquitination and degradation [33]. PI3K is involved in syntheses of HIF-1α by activating its target AKT and a downstream component mammalian target of rapamycin (mTOR) [34]. Subsequently, mTOR phosphorylates eukaryotic translation initiation factor 4E (eIF-4E) binding protein (4E-BP1) leading to enhanced HIF-1α translation [35]. Another pathway that works similarly is the RAS/RAF/MEK/ERK kinase cascade. The ERK can not only phosphorylate 4E-BP1 but also co-activator CBP/p300, thereby stimulating transcriptional activation of HIF-1α target genes [36]. An overview of HIF-1α regulation is shown in Figure 1.

### 2.2. The Role of the HIF-1α Pathway in Cellular Adaptation to Hypoxia

As mentioned, HIF-1α is part of the adaptive response of tumor cells to hypoxia, as it activates up to 200 genes crucial for tumor proliferation and metastasis. For example, HIF-1α induces various pro-angiogenic factors, such as the vascular endothelial growth factor (VEGF), to develop new blood vessels and increase oxygen and nutrient transport to the tumor [37]. HIF-1α also activates several oncogenic growth factors, thereby stimulating tumor metastasis to more oxygenated regions [38]. Furthermore, HIF-1α increases resistance against apoptosis in tumor cells by inhibition of Bid and Bax, which are pro-apoptotic proteins of the Bcl-2 family, as well as by enhancing the expression of apoptosis inhibitors [39,40]. Additionally, HIF-1α is involved in regulating several aspects of cell metabolism.

#### 2.2.1. Glucose Metabolism

Here, HIF-1α mediates the shift in glucose metabolism from oxidative phosphorylation to glycolysis in tumor cells. In doing so, hypoxia stimulates the Warburg effect. As glycolysis produces considerably less ATP per glucose molecule than oxidative phosphorylation, just 2 ATP molecules instead of 36 ATP molecules, a substantial amount of glucose is required in the cell to compensate for this shortage and the high energy needs of the tumor. HIF-1α solves this problem by upregulating the translocation of glucose transporters (GLUTs), such as GLUT1 and GLUT3, to the cell membrane to facilitate greater glucose influx required for glycolysis into the tumor cells [41]. GLUT1 and GLUT3 are both shown to be overexpressed in various types of cancer; however, GLUT1 is overexpressed more often. Interestingly, Mendez et al. discovered that GLUT1 expression in cancer cells is positively linked to its distance from the nearest blood vessel [42]. Additionally, Chen et al. observed overexpression of GLUT1 in the hypoxic regions of breast tumors in a meta-analysis of 12 studies [43]. This again indicates a correlation of GLUT1 expression to hypoxia in breast cancer. Both research groups linked upregulation of glucose transporters to poor prognosis.

Moreover, HIF-1α can enhance glycolytic activity by regulating several glycolytic enzymes in tumor cells. An example is hexokinase, which is responsible for the conversion of glucose to glucose-6-phosphate [44]. This dedicates glucose molecules to the glycolytic cycle, as the cell cannot export glucose-6-phosphate through GLUTs. Likewise, HIF-1α inhibits expression of pyruvate dehydrogenase (PDH), which regulates the conversion of pyruvate to acetyl CoA through transcription of pyruvate dehydrogenase kinase 1 (PDK1) [44]. Consequently, the flux through the tricarboxylic acid (TCA) cycle is suppressed, leading to a greater conversion of pyruvate to lactate, as observed in the Warburg effect. The HIF-1α stimulates this conversion even further through upregulation of lactate dehydrogenase (LDH), which catalyzes the reaction. Elevated levels of LDH are observed in many types of cancer including breast cancer [45]. Twenty years ago, studies already linked elevated LDH levels to poor prognosis in breast cancer patients, as it resulted in an enhanced risk of cancer recurrence and death [45].

#### 2.2.2. Lipid Metabolism

Due to the Warburg effect induced by hypoxia, nutrients normally used in oxidative phosphorylation for ATP production now remain available for lipid and amino acid synthesis to fuel tumor growth and survival. Recent studies indicate that HIF-1α is an important regulator of the lipid metabolism as well [46,47]. The lipid metabolism provides an alternative energy source when glucose levels are low [48]. The lipid metabolism starts with the conversion of triglycerides to glycerol and fatty acids by lipoprotein lipase (LPL). The LPL protein levels are often elevated in breast cancer [49]. Glycerol, also known as dihydroxyacetone phosphate, can directly enter the glycolysis pathway. Fatty acids are first converted to acetyl CoA through β-oxidation and, subsequently, enter the TCA cycle. Under hypoxic conditions, HIF-1α stimulates fatty acid uptake by upregulating fatty acid binding proteins (FABPs), specifically FABP3, FABP7, and FABP4 [50]. Additionally, HIF-1α decreases fatty acid β-oxidation though the suppression of acyl CoA dehydrogenases, thereby again preventing acetyl CoA entry in the TCA cycle [51]. On the other hand, production of acetyl-CoA is enhanced by hypoxia through the upregulation of acetyl-CoA synthase 2 (ACSS2) [52]. The ACSS2 is responsible for the conversion of acetate to acetyl-CoA. Due to inhibited entry in the TCA cycle, this acetyl-CoA is instead used for synthesis of cell membrane phospholipids, among other things. The ACSS2 expression is also linked to increased tumor aggressiveness in breast cancer patients [53]. Overall, hypoxia-induced lipid metabolism reprogramming results in fatty acid accumulation, which promotes tumor growth and survival upon reoxidation [54]. However, the exact role of HIF-1α in tumor lipid metabolism is not well understood at this time and is likely cancer type specific. Therefore, more research is warranted to understand its importance in breast cancer progression.

#### 2.2.3. Amino Acid Metabolism

Another alternative energy source is provided by amino acid metabolism. Due to the importance of hypoxia in glucose and lipid metabolisms, it is likely that is plays a role here as well. The most abundant amino acid in the circulation is glutamine [55]. Glutamine can be used as a building block for protein synthesis once taken up by the cell. However, it is primarily used as a source for cellular energy production. First, glutamine is converted to glutamate by the enzyme glutaminase. Afterwards, glutamate is converted to either α-ketoglutarate, a substrate of the TCA cycle, or pyruvate. Conversion of glutamate to pyruvate is called glutaminolysis and happens at high rates in cancer cells. The protein c-Myc has been shown to stimulate glutaminolysis in multiple ways [56]. For example, c-Myc upregulates glutamine transporters and, thus, glutamine uptake. Additionally, c-Myc increases the expression of glutaminase 1 (GLS1). The role of hypoxia in amino acid metabolism is complex. HIF-1α downregulates c-Myc expression while HIF-2α promotes c-Myc activity [57,58]. A possible explanation for this downregulation is the fact that c-Myc is also sensitizes cells to hypoxia-induced apoptosis. However, upregulation of c-Myc could help in maintaining a functioning TCA cycle in spite of the Warburg effect and provide nutrients. Overall, HIF-1α is one of the key components in the adaption of tumor cells to survive under hypoxic conditions.

#### 2.2.4. Therapeutic Resistance

All these altered mechanisms of cell metabolism make it possible for the tumor to persevere under hypoxic conditions. Besides obtaining protection against the hypoxic microenvironment, the tumor also becomes less susceptible to treatment in it. Anticancer agents are typically large molecules that cannot reach the tumor easily due to the disordered vasculature of the tumor. Additionally, the hypoxic cells in the tumor are present in the outer regions and far from the blood supply. Consequently, the anticancer agents are not delivered to the hypoxic cells enough to expose them to a lethal dose [59]. Furthermore, there are several studies that indicate hypoxic tumors develop resistance to radiotherapy [60,61]. One possible explanation is related to the repair of DNA breaks induced by radiotherapy. In well-oxygenated cells, the oxygen reacts with the broken DNA strands, which creates stable peroxides that are not easily repaired. As this does not happen in hypoxic cells, they may repair the damage more efficiently and survive radiotherapy.

## 3. ECM in Breast Cancer

The ECM is another component in the tumor microenvironment that influences tumor progression. It consists of circa 300 proteins that are involved in tissue development, function, and homeostasis [62]. The ECM is composed of locally secreted fibrous proteins and proteoglycans in a complex meshwork that provides the structural framework for the majority of tissues [63]. Both biophysical and biochemical signals provided by the ECM are important in regulating cell proliferation, differentiation, and survival. Not surprisingly, the ECM also plays a crucial role in tumor invasion and metastasis [64,65].

### 3.1. Stiffness and Topography of the ECM

As the structure of the ECM is dynamic, interferences in this network can disrupt cell and tissue homeostasis and lead to diseases, such as fibrosis. This is caused by excessive accumulation of the ECM due to a wound-healing response upon chronic injury including cancer [66]. The sustained activation of local fibroblasts to myofibroblast is an important part of this process as they secrete and assemble the ECM. This overproduction of ECM eventually leads to the formation of a fibrotic lesion. Tumors are usually fibrotic and comprise an increased and atypical ECM deposition that increasingly stiffens the stroma [67]. Consequently, the biophysical properties, such as stiffness, topography, porosity, and solubility of tumor-associated ECM, vary profoundly from that of regular tissue stroma. The biophysical properties of the ECM are especially significant in breast cancer initiation as increased breast density observed by mammography enhances the possibility of developing breast cancer between 4- and 6-fold [68,69]. This increased breast density is often associated with an enhanced deposition of fibrous ECM protein collagen I and III, as well as proteolysis of collagen IV [70].

Collagens frequently accumulate in larger quantities early on in cancer development [71]. They can amount up to 90% of the ECM components and are, therefore, the most dominant ECM component [72]. The extracellular processing of collagens starts with cleavage of procollagen in the ECM by proteinases. There are currently 42 different procollagen α-chains identified in vertebrates, which generate 28 varying collagen subtypes [73,74]. Afterwards, intermolecular cross-links are formed between collagens and elastin by enzymes, such as lysyl oxidases (LOX) and lysyl hydroxylases. This changes the structure, elasticity, and strength of the ECM. While collagen fibers are often disordered in ordinary tissues, they are highly aligned and organized in tumor-associated ECM. Additionally, the fibers are frequently positioned perpendicular to the tumor, thereby allowing tumor cells to migrate via the fibers into surrounding tissue, leading to metastasis [75]. Moreover, Barker et al. demonstrated elevated amounts of LOX in aggressive breast tumors and indicated that it is a marker for invasion, metastasis, and a poor patient outcome [76]. This was further confirmed by Pickup et al. who showed that inhibition of LOX activity in mouse models suppresses tumor metastasis [77].

As collagens play an important role in ECM dynamics, they are used in the clinic as markers of breast cancer progression. Three levels of tumor-associated collagen signatures (TACS) are defined. The TACS-1 level is during early cancer development, when there is an increased amount of collagen in the primary tumor. In TACS-2, the tumor increases in size and the collagen fibers align parallel to the tumor. Once the collagen fibers are aligned perpendicular, the highest risk signature, namely TACS-3, is appointed. At that time, the tumor is very stiff, which increases the risk of mortality in women with invasive breast cancer [75]. Taken together, changes in stiffness and topography of the tumor ECM associate with, and are believed to contribute to, cancer progression.

### 3.2. Mechanotransduction

Cells can react to changes in the tumor microenvironment through mechanotransduction via mechanosensitive structures, such as cell–cell or cell–ECM adhesions. This mechanism involves specialized mechanosensitive ion channels and adhesion receptors, such as integrins, that transmit signals into the cells via a linked cytoskeletal–molecular motor network. Upon receiving the signal, the cell can alter their shape, architecture, and cellular tension, which eventually leads to remodeling of the ECM. Notably, mechanotransduction via cell–cell adhesions usually results in cell growth inhibition, while mechanotransduction via cell–ECM adhesions stimulates cell growth.

The most well-known cell-ECM mechanotransduction mechanism starts with ECM-dependent integrin activation and clustering. Integrin-associated adhesion plaque protein focal adhesion kinase (FAK) subsequently stimulates the formation of focal adhesions and Rho–ROCK-dependent actin remodeling. Moreover, FAK also activates key signaling pathways for cell growth, survival, migration, and invasion. An example is the PI3K/AKT/mTOR pathway necessary for cellular metabolism, which is implicated in tumor aggression and drug resistance. In addition, ECM stiffness following mechanotransduction has been shown to decrease the expression of PI3K inhibitor and the tumor suppressor PTEN, leading to increased tumor aggressiveness [78]. Additionally, breast tumors with increased ECM stiffness often have increased levels of integrins and focal adhesions and, therefore, elevated mechanotransduction [79,80]. An overview of these mechanotransduction pathways is visible in Figure 2.

#### YAP/TAZ Signaling

Linked to these mechanotransduction pathways is YAP/TAZ signaling. Yes-associated protein (YAP) and transcriptional co-activator with PDZ-binding motif (TAZ) are transcriptional regulators of the Hippo tumor suppressor pathway activated in most solid tumors. Although YAP/TAZ seems dispensable for homeostasis in normal tissue, it is essential in tissue repair. This distinction in the necessity of YAP/TAZ in normal and tumor tissue makes it an interesting target for cancer treatment. In addition, elevated YAP/TAZ signaling has been shown to be crucial for breast cancer initiation, progression, metastasis, and drug resistance, and predicts poor patient outcome [81].

A key feature of YAP/TAZ is their role as transducers of extracellular cues. They react to shifts in the microenvironment and cell structure and, thus, convert changes at the tissue level to intracellular biochemical signals [82]. Here, YAP/TAZ signaling is generally inhibited by the upstream proteins in the Hippo pathway. The Hippo pathway starts with phosphorylation of large tumor suppressor homologue 1 (LATS1) and LATS2 by mammalian STE20-like protein kinase 1 (MST1), MST2, and their cofactor scaffold adaptor protein Salvador (SAV). Then, LATS1 and LATS2 phosphorylate YAP and TAZ, leading to their proteasomal degradation [83]. Extracellular cues cause inactivation of the Hippo pathway, allowing YAP/TAZ to translocate to the nucleus and cooperate with several promotor-specific transcription factors, primarily TEA domain family members (TEAD), to regulate the expression of more than 100 target genes implicated in cell survival, proliferation, migration, and angiogenesis [82,84].

The ability of YAP/TAZ to influence tumor progression can be attributed to its function in promoting cell proliferation through multiple mechanisms. They are directly involved in cell replication, as they activate target genes engaged in the regulation of S-phase entry and mitosis [85]. In addition, elevated YAP/TAZ levels can overcome anoikis and mitochondrial-induced apoptosis by upregulating anti-apoptotic Bcl2-family members [86,87]. Furthermore, YAP/TAZ have been implicated in cancer stem cell (CSC) development.

Here, YAP/TAZ activates several oncogenes of the CCN family, such as CCN family member 1 (CCN1) and CCN2, also known as connective tissue growth factor (CTGF) and cysteine-rich angiogenic inducer 61 (CYR61), respectively. Both CNN proteins are involved in a range of key biological processes, such as cell proliferation, angiogenesis, wound repair, fibrosis, and tumorigenesis, although mechanistically this is not fully resolved [88]. The CCN proteins are connected to the ECM and are associated with cell adhesion, signaling, and migration. This is supported by the observation that CCN1 and CCN2 both regulate cell adhesion in various cell types. Furthermore, CCN2 is essential for ECM contraction [89]. In addition to promoting cell adhesion, CCN proteins enhance cell survival by preventing apoptosis through interaction with transforming growth factor β (TGF-β) and various surface integrins. In that way, CCN1 and CCN2 mediate the synthesis of ECM components and angiogenesis as well. Their induction of angiogenesis is dependent on the growth factor they interact with. For example, CCN2 can inhibit angiogenesis when interacting with VEGF or stimulate it by interacting with TGF-β. Ultimately, both CCN1 and CCN2 show increased expression in advanced breast cancer, and they increase malignancy [90]. This once more suggests the importance of YAP/TAZ signaling in breast cancer progression.

## 4. Hypoxia Modulates Composition of the ECM

Previously, it was believed that hypoxia and the ECM were separate components in the tumor microenvironment. However recent research has shown that the two factors can interact in the TME.

### 4.1. Hypoxia Influences ECM Deposition

As mentioned earlier, excessive accumulation of the ECM can cause fibrosis and stiffens the ECM. Fibrosis in breast cancer leads to poorer prognosis and added risk of recurrence [91]. Histological studies of human breast tumors assessed by CG Colpaert et al. show that fibrosis is most prevalent in the hypoxic regions within a tumor. They discovered this by immunostaining of HIF-1 target gene carbonic anhydrase IX (CAIX) in a breast cancer cell line [92]. Additionally, CAIX immune reactivity is abundant in highly fibrotic tumors and is, therefore, considered a marker for poor patient outcome.

There are several mechanisms by which hypoxia influences ECM deposition. The first one involves the activation of fibroblasts. Local fibroblasts and myofibroblasts in the ECM are initiated by several proteins emitted by the tumor. One of the most noteworthy is transforming growth factor-β (TGFβ), which is induced by hypoxia [93]. This provides a logical reason for the increased ECM deposition in the hypoxic regions of the tumor. Additional evidence is provided in several studies on dermal, cardiac, and renal fibroblasts that show elevated type I procollagen α1 mRNA levels when cultured in hypoxic environments [94,95,96].

#### 4.1.1. Proline Hydroxylation

There are various post-translational modifications of the procollagen α-chains, such as hydroxylation of proline and lysine residues. Hydroxylation of proline to 4-hydroxyproline is vital in improving the thermal stability of procollagen. It also makes sure the procollagen is folded properly into stable triple helixes to prevent degradation [97]. This process is catalyzed by prolyl 4-hydroxylase (P4H), which is generated by the formation of tetramers of prolyl 4-hydroxylase-α subunit (P4HA) with prolyl 4-hydroxylase-β subunit (P4HB). At this time, three isoforms of P4HA are identified, namely P4HA1, P4HA2, and P4HA3 [98]. Interestingly, HIF-1α is involved in the regulation of P4HA1 and P4HA2 expression in tumor cells and fibroblasts. Inhibition of HIF-1α, P4HA1, or P4HA2 via short hairpin RNAs has been shown to decrease collagen deposition in breast cancer cells and fibroblasts in vitro [99,100]. Additionally, in vivo experiments confirm this observation by proving that inhibiting HIF-1α, P4HA1, or P4HA2 in orthotopic tumors, developed via injection of human breast cancer cells into immunodeficient mice, decreases tissue stiffness and fibrosis [99,101]. The P4HA1 or P4HA2 knockdown also prevented breast cancer metastasis in mice by inhibiting collagen fiber development, thereby making it challenging for tumor cells to migrate into surrounding tissue [99,101]. Altogether, this suggests hypoxia indeed has a substantial role in ECM deposition through its transcription factor HIF-1α. Notably, HIF-2α expression seemed to have no effect on ECM deposition related to proline hydroxylation in breast cancer cells.

#### 4.1.2. Lysine Hydroxylation

Lysine hydroxylation is another important post-translational modification of procollagen. Lysine hydroxylation is utilized to form collagen crosslinks that display enhanced stability, unlike non-hydroxylated lysine residues [102]. Consequently, it increases tissue stiffness and is a general phenomenon in fibrosis. Collagen lysine hydroxylation is facilitated by enzymes encoded by three different procollagen-lysine 2-oxyglutarate 5-dioxygenase genes, namely PLOD1, PLOD2, and PLOD3 [101]. Here, PLOD1 hydroxylates lysine in the α-helical or central domain of procollagens, while PLOD2 is responsible for hydroxylation of lysine residues in the telopeptide of procollagens. Both are transcriptionally activated by HIF-1α. The exact function of PLOD3 is currently unknown. Remarkably, only PLOD2 activation is able to change collagen cross-linking patterns [103]. While PLOD2 depletion did not inhibit collagen deposition in breast cancer in vitro or in vivo, as in the case of P4HAs, it did decrease tumor stiffness. Additionally, PLOD2 knockdown substantially reduced metastasis of breast cancer cells in mice [104]. Hence, PLOD2 activation induced by HIF-1α is especially critical to ultimately increase fibrillar collagen formation and tumor stiffness.

#### 4.1.3. Lysyl Oxidases

Besides influencing intracellular collagen-modifying enzymes, hypoxia also plays an important role extracellularly. Through hydroxylation of proline and lysine residues, procollagen α-chains form a triple helix and are secreted to the ECM. There, further collagen cross-linking between collagens and elastin is mediated by LOX enzymes. These target genes of the HIF pathway, namely LOX, LOX-like protein 2 (LOXL2), and LOXL4, are strongly induced by hypoxia [105,106]. In breast cancer, LOX enzymes are upregulated as a response to hypoxia to various degrees depending on the tumor type [107]. LOX upregulation in that case is mediated by HIF-1α. A study in which MDA-MD-231 subclones were transplanted in mice showed decreased LOX, decreased collagen-cross-linking, and a softer ECM, with reduced metastasis initiation in clones with decreased HIF-1α levels [108].

In sum, hypoxia directly regulates ECM composition via multiple enzymes and, as such, P4HA1, P4HA2, PLOD2, and LOX enzymes could be used as biomarkers in breast cancer progression.

### 4.2. Hypoxia Leads to ECM Degradation

Besides stiffening the ECM through collagen modifications and cross-linking, hypoxia also has an effect on collagen degradation. This form of ECM remodeling is mediated by numerous proteinases, including members of the matrix metalloproteinases (MMPs) family, which are frequently overexpressed in tumors [109]. The MMPs are zinc-dependent enzymes that are collectively capable of degrading ECM components and extracellular basement membrane. As barriers surrounding the tumor are degraded, MMPs have been proposed to promote tumor metastasis. There are several subcategories of MMPs, such as collagenases, stromelysins, gelatinases, and cell membrane bound MMPs, which all have distinct substrates.

In breast cancer, hypoxia mainly upregulates the activity of collagenases. More specifically, type IV collagen-degrading enzymes MMP-2 and MMP-9 show increased expression through a HIF-1α dependent mechanism in in vitro breast carcinoma cell lines [110,111]. Additionally, hypoxia increases the expression of membrane type 1 MMP (MT1-MMP), also known as MMP-14. The MT1-MMP has a number of functions, including converting pro-MMP-2 to MMP-2, and in the cleavage of several ECM components, such as collagen type I, II, and III. Interestingly, MT1-MMP is upregulated by both HIF-1α and HIF-2α expression [110,112]. Hypoxia also indirectly increases MMP expression by promoting accumulation of macrophages in the primary tumor by inducing various growth factors [113]. These macrophages produce MMPs, stimulate additional growth factor production, which attracts more macrophages, and promote tumor intravasation [114,115].

Furthermore, hypoxia results in increased expression of plasminogen activator urokinase receptor (PLAUR) via HIF-1α expression in breast cancer [116]. As with MMPs, PLAUR activation also leads to the degradation of several ECM components, including the basement membrane. In addition, PLAUR activation causes alterations in the interaction of integrins with the ECM, thereby allowing tumor invasion. Taken together, hypoxic signaling engages multiple mechanisms that contribute to ECM remodeling, ultimately increasing breast cancer aggressiveness.

## 5. Crosstalk between Hypoxia and the ECM

A growing amount of research supports the hypothesis that hypoxia and the ECM cooperate to influence breast cancer progression. This is mostly evident in their roles in altering cell metabolism and stimulating tumor metastasis. However, crosstalk between hypoxia and the ECM is still poorly understood.

### 5.1. The ECM and Hypoxia Alter Glycolytic Cell Metabolism Together

The impression that the ECM can influence cell behavior was reinvigorated in the field of mechanotransduction. The cell–ECM mechanotransduction mechanism, induced by hypoxia, among other factors, leads to the activation of multiple downstream signaling pathways for tumor growth, survival, migration, and invasion. This includes the upregulation of glycolytic enzymes previously mentioned as a result of hypoxia.

#### 5.1.1. Glycolytic Enzymes

A good example of a glycolytic enzyme regulated by the ECM is phosphofructokinase (PFK). Here, PFK catalyzes conversion of fructose-6-phosphate into fructose 1,6-bisphosphate and ADP via ATP-dependent phosphorylation. It is, therefore, a key step in the regulation of glycolysis. Normally, the E3 ubiquitin ligase tripartite motif containing protein 21 (TRIM21) degrades PFK, thereby inhibiting glycolysis. This is most likely a tumor suppressive mechanism in place to reduce cell proliferation and prevent overgrowth. Conversely, TRIM21 is trapped by actin bundling and stress fiber formation on a stiffer ECM typical in a tumor and, as such, cannot degrade PFK [117].

Another example is the glycolytic enzyme aldolase. It reversibly catalyzes the conversion of fructose 1,6-bisphosphate to dihydroxyacetone phosphate (DHAP) and glyceraldehyde 3-phosphate (G3P). Aldolase is usually bound to the actin filaments of the cytoskeleton [118]. However, Ras-related C3 botulinum toxin substrate 1 (RAC1), which is a member of the Rho family of GTPases, can release aldolase through actin remodeling [119]. This RAC1 activity is normally inhibited on a soft ECM, thereby also suppressing glycolysis and perhaps supporting the TRIM21 tumor suppressive mechanism explained above. Interestingly, RAC1 is activated by PI3K signaling, which is activated on a stiff ECM and regulates HIF-1α expression, thereby providing an additional link between the ECM and hypoxia. Moreover, RAC1 is often overexpressed in breast cancer and is associated with multi-drug resistance [120].

#### 5.1.2. Intracellular pH

The ECM, along with hypoxia, makes the glycolytic metabolism more efficient by regulating intracellular pH. Due to hypoxia, cells switch towards aerobic glycolysis, which results in increased lactate production. This excess lactate causes a decrease in intracellular pH which would consequently reduce glycolysis. Therefore, it must be rapidly secreted by the cell to prevent the acidification of the cell. The transport of lactate is mediated by the monocarboxylate co-transporters MCT1 and MCT4, as well as sodium-hydrogen antiporter 1 (NHE1) [121]. Here, MCT4 expression is upregulated by both a stiffer ECM and hypoxia in breast cancer through transcriptional activation of MCT4 by the binding of HIF-1α to hypoxia response elements [122,123]. Additionally, overexpression of MCT4 is considered a marker for poor prognosis in cancer. Although there is currently no indication that the ECM stiffness or hypoxia regulate MCT1 expression, MCT1 is regulated by several transcription factors including c-Myc. NHE1 is regulated by the ECM as it is a downstream target of the Rho/ROCK mechanotransduction signaling pathway [124]. NHE1 overexpression has also been shown to lower cell–cell adhesion strength, which reduces the metastatic ability of tumor cells, while increasing cell–ECM interaction [125]. The lactate that is now secreted by the MCTs and NHE1 acidifies the tumor microenvironment, thereby allowing degradation of the ECM basement membrane, escape of immune surveillance, and tumor growth and metastasis [126].

All in all, the ECM and hypoxia alter glycolytic cell metabolism in multiple ways. They mutually enhance aerobic glycolysis through upregulation of glucose transport, glycolytic enzymes, and by regulating intracellular pH.

### 5.2. Both Mechanotransduction and Hypoxia Affect Lipid Metabolism

Mechanotransduction, similarly to hypoxia, also affects the lipid metabolism of the tumor via various pathways. For example, ECM stiffness regulates low-density lipoprotein receptor (LDLR), fatty acid transporter CD36, lipoprotein lipase (LPL), and sterol regulatory element-binding proteins (SREBP). The SREBPs are particularly important, as they regulate most of the enzymes involved in the lipid synthesis. Overexpression of SREBP has been shown to enhance tumor growth and survival [127]. The ECM stiffness especially regulates SREBP1 and SREBP2 via multiple already described pathways. First, the PI3K/AKT pathway activated by integrin signaling due to the stiffening of the ECM can activate CREB regulated transcription coactivator 2 (CRTC2) [128]. CRTC2 inhibits proteasomal degradation of SREBP1 and SREBP2, thereby upregulating their activity. Additionally, AKT promotes the uptake of LDL through the LDLR [129]. Secondly, actin remodeling through Rho/ROCK signaling regulates LPL activity through its translocation to the cell surface [130]. LPL breaks down VLDL and chylomicrons, thereby enhancing free fatty acid levels. In spite of this, actin remodeling also negatively regulates the processing of SREBP in the Golgi apparatus, thus, inhibiting its activation. Third, YAP/TAZ mechanosignaling has also been shown to upregulate lipid synthesis through SREBP expression. Moreover, YAP knockdown drastically decreased lipogenesis, while overexpression of YAP enhanced lipogenesis [131]. This suggests that the ECM affects lipid metabolism in much the same way hypoxia does, as both stimulate fatty acid uptake and synthesis, resulting in fatty acid accumulation in the cell. This provides the tumor with additional energy for growth and metastasis.

### 5.3. ECM Stiffness Like Hypoxia Regulates Amino Acid Metabolism

Furthermore, mechanotransduction pathways, such as integrin-activated FAK/PI3K/AKT and YAP/TAZ signaling also regulate the amino acid metabolism in multiple ways. The amino acid metabolism is important for tumor growth, as it is an alternative energy source and provides building blocks for protein synthesis. Concisely, ECM stiffness influences the amino acid metabolism by affecting expression of amino acid synthesis enzymes, glutamine catabolism, and amino acid transport. For example, FAK/PI3K/AKT signaling are involved in amino acid metabolism by the AKT-mediated activation of amino acid transporter SLC6A19 and the upregulation of CD98, which stimulates amino acid uptake in the cell [129,132].

Additionally, YAP/TAZ has a major role in amino acid biosynthesis. Upregulation of YAP/TAZ due to a stiff ECM alters the expression of several enzymes, such as phosphoserine aminotransferase 1 (PSAT1), phosphoserine phosphatase (PSPH), and phosphoglycerate dehydrogenase (PHGDH), that are involved in the biosynthesis of the amino acid serine and serine hydroxy methyltransferase 2 (SHMT2) [133]. These, in turn, are relevant to glycine and aspartate aminotransferase syntheses, which can catalyze production of aspartate. Aspartate is a precursor for some essential amino acids. This mechanism is supported by in vitro and in vivo experiments consisting of knockout of upstream genes of YAP/TAZ signaling which resulted in inhibition of serine and glycine production due to reduced PSAT1, PSPH, and SHMT2 expression [134]. Additionally, the YAP/TAZ pathway enhances glutaminase transcription in cancer cells [135]. Glutamine, like aspartate, serves as precursor for amino acids and is involved in their synthesis. Moreover, it contributes to the TCA cycle to provide cellular energy. The complex regulation of glutamine by hypoxia was described earlier and again indicates crosstalk.

Furthermore, YAP/TAZ is involved in amino acid transport. YAP/TAZ expression results in upregulation of transporters SLC1A3, SLCA5, SLC38A1, SLC7A5, and SLC3A2 [135,136,137,138]. Respectively, they are aspartic and glutamic acid, neutral, glutamine, leucine, and neutral transporters. Based on a Gene Expression Omnibus (GEO) database, SLC1A5 overexpression is associated with poor patient prognosis in breast cancer [136].

All in all, these findings suggest a role for mechanotransduction similar to hypoxia in the synthesis and transport of amino acids in the amino acid metabolism, mainly through YAP/TAZ signaling. Thus, YAP/TAZ may link the ECM and cell metabolism and regulate tumor growth, survival, and aggressiveness in breast cancer.

### 5.4. YAP/TAZ Is the Missing Link between Tumor Microenvironment and Glycolytic Cell Metabolism

As explained, YAP/TAZ is regulated through extracellular signals, such as ECM stiffness, which results in increased tumor aggressiveness. However, intracellular signals, such as cell metabolism, are subject to regulation by hypoxia and the ECM, and can regulate YAP/TAZ signaling as well, i.e., when cellular energy stress detected by AMP-activated protein kinase (AMPK) inhibits YAP activation. AMPK maintains cellular energy homeostasis by inhibiting energy consuming processes, such as cell growth, and by activating catabolic pathways that produce ATP upon sensing low energy levels [139]. For example, AMPK can increase glucose uptake via GLUT1 and GLUT4 [140,141]. It has also been implicated that AMPK plays an essential part in stimulating glycolysis under hypoxic conditions [142]. An overview of how YAP/TAZ becomes a link between hypoxia and the ECM is presented in Figure 3.

#### 5.4.1. YAP/TAZ Is Regulated by Both the Microenvironment and Glycolytic Cell Metabolism through AMPK

AMPK uses two different mechanisms to inhibit YAP signaling. First, AMPK can directly phosphorylate YAP at Serine 94 [143]. This prevents the complex formation of YAP/TAZ with TEAD, as Serine 94 in YAP usually forms a hydrogen bond with Tyrosine 406 in TEAD1. As YAP has no DNA binding domain itself, the disruption of the YAP–TEAD complex inhibits activation of its target genes. Notably, AMPK is able to phosphorylate YAP at other sites as well, including Serine 61, Serine 366, and Serine 463 [143,144]. Phosphorylation of those sites does not distort the binding of YAP with TEAD but does inhibit transcriptional activation. The mechanism behind this is currently not understood. TAZ has a similar Serine binding with TEAD as it does with YAP, which suggests that AMPK could regulate TAZ likewise. However, this has not been explicitly researched yet.

The second mechanism in which AMPK can inhibit YAP/TAZ is indirect, as AMPK stabilizes tight junction protein angiomotin (AMOT) through phosphorylation [145]. Here, AMOT, AMOT-like 1(AMOTL1), and AMOTL2 can inhibit YAP by multiple mechanisms. One of these involves binding to YAP and sequestering it from the nuclei. Another is by binding to LATS upstream in the Hippo pathway and enhancing its activation [146]. The last involves promoting YAP ubiquitination and, thus, decreasing its stability [146].

The ability of YAP/TAZ to respond to both intracellular and extracellular signals proves that it is a clear link between the microenvironment and cell metabolism in a tumor. Bearing in mind the oncogenic role of YAP/TAZ in stimulating tumor progression and metastasis, it is not entirely unexpected that its activity is inhibited by cellular energy stress. Tumors need a lot of energy for cell growth, and proliferation should not continue with limited energy levels. Due to the increased ATP production via the Warburg effect in aggressive hypoxic tumors, AMPK levels are low, resulting in increased YAP/TAZ. Overall, both mechanotransduction via increased ECM stiffness and altered cell metabolism through hypoxia stimulate YAP/TAZ signaling.

#### 5.4.2. YAP/TAZ Regulates Both the Tumor Microenvironment and Glycolytic Cell Metabolism

It must be noted that YAP/TAZ signaling is not only regulated by the tumor microenvironment and cell metabolism, but it also regulates it. For instance, one of the target genes of YAP regulates GLUT3 [147]. Thus, YAP promotes glycolysis via increased glucose uptake and indirectly stimulates its own expression. Another interesting case is its target gene c-Myc, which was mentioned before to be both positively and negatively regulated by hypoxia.

Indeed, c-Myc is a key transcription factor that regulates around 15% of all human genes [148]. Consequently, c-Myc is a critical factor in cell growth, metabolism, proliferation and survival. As these functions are all important in tumor progression, c-Myc is a strong promoter of tumorigenesis and is often deregulated and constitutively expressed in various cancers. In breast cancer, c-Myc deregulation is frequently associated with aggressive tumors and poor patient outcomes [149].

Primarily, c-Myc is established as a driver of proliferation. For example, it causes genomic instability in tumors by creating DNA replication stress via multiple mechanisms. Thus, c-Myc promotes factors that stimulate S-phase entry and cell growth, and downregulates p21 [148,150]. In addition, c-Myc is directly involved in DNA replication by interacting with the pre-replicative complex and localizing to initial sites of DNA synthesis [151].

Moreover, c-Myc is an important regulator of glycolysis through its target genes and, thus, directly contributes to the Warburg effect. One example is the target gene lactate dehydrogenase A (LDHA), which plays a role in converting pyruvate to lactate in the glycolytic pathway [152]. Other genes include GLUT1, hexokinase 2 (HK2), phosphofructokinase (PFKM), and enolase 1 (ENO1). Under hypoxic conditions, c-Myc also works together with HIF-1 to stimulate PDK1, thereby favoring the conversion of glucose to lactate [153]. The effect c-Myc has on glycolysis has been proven in studies with transgenic mice [154].

This further indicates that YAP/TAZ signaling is an important linker of hypoxia and the ECM in the tumor microenvironment on breast cancer progression.

## 6. Conclusions

Hypoxia and the ECM are critical components of the TME driving breast cancer progression. Tumor cells mainly adapt to hypoxic conditions through activation of the transcription factor HIF-1α to alter cell metabolism. Tumors remodel the ECM, and the resulting altered ECM stiffness and topography contributes to invasion and metastasis. Hypoxic signaling engages multiple mechanisms that directly contribute to ECM remodeling, ultimately increasing breast cancer aggressiveness. There is evidence pointing to crosstalk between hypoxia and the ECM, in particular in deregulating cell metabolism, which is regarded as a hallmark of aggressive cancers. First, hypoxia and the ECM cooperate to alter different aspects of cell metabolism. They mutually enhance aerobic glycolysis through upregulation of glucose transport, glycolytic enzymes, and by regulating intracellular pH. Second, they alter lipid and amino acid metabolism by stimulating lipid and amino acid uptake and synthesis, thereby providing the tumor with additional energy for growth and metastasis. Third, YAP/TAZ signaling is regulated by the TME and cell metabolism and, vice versa, regulates cell metabolism primarily through its target c-Myc. Here, we largely concentrated on breast cancer, but the guiding principles discussed in this review likely extend to other cancer types as well.

## Figures and Tables

**Figure 1 genes-13-01585-f001:**
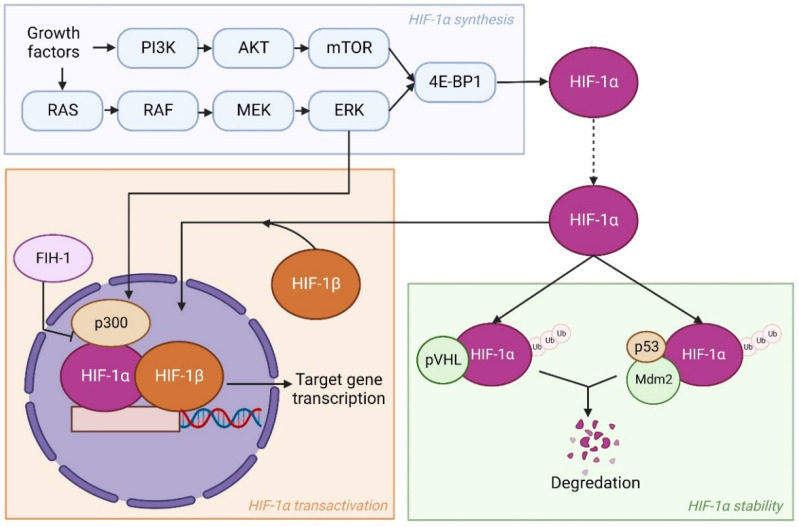
HIF-1α regulation by altering HIF-1α synthesis, stability, or transactivation. The HIF-1α synthesis is mediated by the PI3K-AKT pathway and the RAS/RAF/MEK/ERK kinase cascade upon activation by growth factors. Afterwards, HIF-1α is either degraded via pVHL or p53/Mdm2 related ubiquitination, or it translocates to the nucleus. There, it forms a heterodimer with HIF-1β and is coactivated by p300 leading to transcriptional activation of target genes. This can be enhanced by ERK expression or inhibited by FIH-1. Created with BioRender.com, accessed on VN24CTMQMS, 9 July 2022.

**Figure 2 genes-13-01585-f002:**
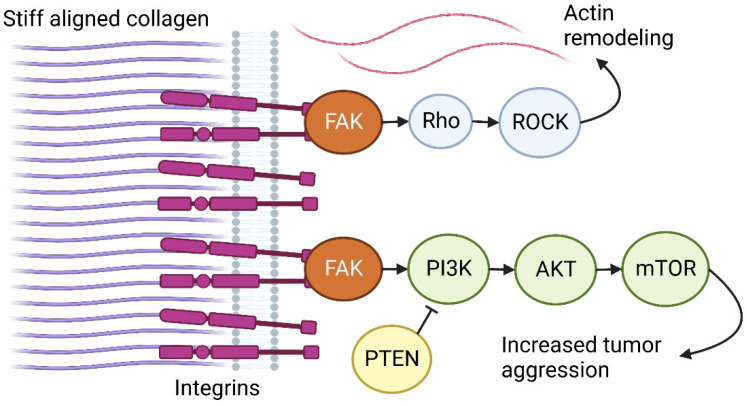
Mechanotransduction. Increased ECM stiffness activates integrin-dependent focal adhesion formation. Consequently, several pathways are activated, such as Rho/ROCK and PI3K/AKT/mTOR, which cause actin remodeling and increased tumor aggression. Created with BioRender.com, accessed on QE24CTMVV9, 9 July 2022.

**Figure 3 genes-13-01585-f003:**
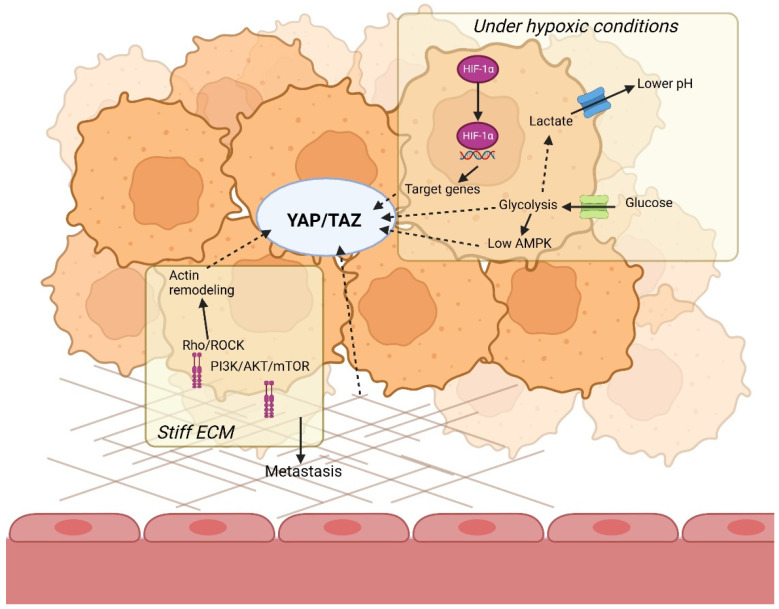
YAP/TAZ as the missing link between tumor microenvironment and glycolytic cell metabolism. Thr YAP/TAZ is stimulated by the stiffening of the ECM through multiple mechanotransduction pathways. It is also positively regulated by the glycolytic cell metabolism and negatively by AMPK. In addition, it is connected to downstream targets of HIF-1α. Created with BioRender.com, accessed on BW24CTMYPN, 9 July 2022.

## Data Availability

Not applicable.

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
