# Peer review of "Crosstalk between Hypoxia and Extracellular Matrix in the Tumor Microenvironment in Breast Cancer"

_genes, 2022, doi:10.3390/genes13091585_

Round 1

Reviewer 1 Report

This is a readable and well structured review of how hypoxia and ECM factors interact in the progression of breast cancer. It is generally comprehensive and scientifically sound. Overall I would possibly suggest that focussing on breast cancer is irrelevant for a large part of this - most of the concepts are relevant to other cancers and that is reflected in a lot of the sources used. I also think the figures could be a bit clearer. Overall though I think this would be a reasonable resource in the field and there is a well structured argument linking the two concepts together.

Some points to consider:

- line 47 would benefit from more clarity over how pH allows tumour invasion and suppression of immunity. At very least needs references.

- line 57 I don't think "learned" is appropriate when we are discussing adaptation of cancer cells.

- line 77 supporting might be a better word than confirming.

- line 94 is 48% AA similarity really "very" similar?

- line  146 "several pathways" is vague and would benefit from some examples.

- line 163 some clarity about why glucose cannot leave the cell anymore would be beneficial to readers (and could be more technically precise - cellular export?).

- line 200 and various points onwards refers to "the" amino acid metabolism where "the" is redundant.

- line 269 - "they are a marker" should be "it is a marker".

- line 311 is YAP/TAZ pathway really "another way" of mechanotransduction? It is linked to integrin signalling, this isn't accurate.

- line 436 would benefit from clarifying the collective action of MMPs in degrading ECM, to reinforce their distinct substrates but collective activity.

- line 445-6 not clear how macrophages are inducing MMP activation, is it stimulation (how?) or synthesis of MMPs?

line 478 would degrade be a better phrasing than "cannot break down"?

line 492 technically imprecise to say the ECM "works with" hypoxia, needs rephrasing.

- line 587 is stiffening of ECM "signalling"? Or does signalling result from stiffening of ECM?

Reviewer 2 Report

This an excellent, well organized and presented review concerning the background and role and possible therapeutic targeting of the cross-talk between hypoxia and the extracellular matrix of the tumor microenvironment in directing breast tumor development, progression and aggressiveness. It was easy to read and understand and will be an important review for both young researchers and physians and for those who have been in the field already for some time.

Author Response

No edits were made based on this review.